# Analysis of Antidiabetic Activity of Squalene via In Silico and In Vivo Assay

**DOI:** 10.3390/molecules28093783

**Published:** 2023-04-27

**Authors:** Tri Widyawati, Rony Abdi Syahputra, Siti Syarifah, Imam Bagus Sumantri

**Affiliations:** 1Department of Pharmacology and Therapeutic, Faculty of Medicine, Universitas Sumatera Utara, Medan 20155, Sumatera Utara, Indonesia; 2Department of Pharmacology, Faculty of Pharmacy, Universitas Sumatera Utara, Medan 20155, Sumatera Utara, Indonesia; 3Department of Pharmaceutical Biology, Faculty of Pharmacy, Universitas Sumatera Utara, Medan 20155, Sumatera Utara, Indonesia

**Keywords:** diabetic mellitus, in silico, in vivo, squalene

## Abstract

Squalene has been tested widely in pharmacological activity including anticancer, antiinflammatory, antioxidant, and antidiabetic properties. This study aims to examine antidiabetic activity of squalene in silico and in vivo models. In the in silico model, the PASS server was used to evaluate squalene antidiabetic properties. Meanwhile, the in vivo model was conducted on a Type 2 Diabetes Mellitus (T2DM) with the rats separated into three groups. These include squalene (160 mg/kgbw), metformin (45 mg/kgbw), and diabetic control (DC) (aquades 10 mL/kgbw) administered once daily for 14 days. Fasting Blood Glucose Level (FBGL), Dipeptidyl Peptidase IV (DPPIV), leptin, and Superoxide Dismutase (SOD) activity were measured to analysis antidiabetic and antioxidant activity. Additionally, the pancreas was analysed through histopathology to examine the islet cell. The results showed that in silico analysis supported squalene antidiabetic potential. In vivo experiment demonstrated that squalene decreased FBGL levels to 134.40 ± 16.95 mg/dL. The highest DPPIV level was in diabetic control- (61.26 ± 15.06 ng/mL), while squalene group showed the lowest level (44.09 ± 5.29 ng/mL). Both metformin and squalene groups showed minor pancreatic rupture on histopathology. Leptin levels were significantly higher (*p* < 0.05) in diabetic control group (15.39 ± 1.77 ng/mL) than both squalene- (13.86 ± 0.47 ng/mL) and metformin-treated groups (9.22 ± 0.84 ng/mL). SOD activity were higher in both squalene- and metformin-treated group, particularly 22.42 ± 0.27 U/mL and 22.81 ± 0.08 U/mL than in diabetic control (21.88 ± 0.97 U/mL). In conclusion, in silico and in vivo experiments provide evidence of squalene antidiabetic and antioxidant properties.

## 1. Introduction

Diabetes Mellitus (DM) is a metabolic chronic disease on a global scale that characterized by elevated Blood Glucose Level (BGL). This condition is associated with disruptions in carbohydrate, lipid, and protein metabolism due to insulin deficiency. Insulin deficiency can arise from either impaired insulin production by beta Langerhans cells in the pancreas gland or a lack of insulin responsiveness in the body cells [1,2,3]. In addition to insulin, the adipocyte hormone leptin also plays a regulatory role in food intake and energy expenditure [2], which are also relevant to the development of DM.

Globally, DM affects a staggering 347 million people, and it is responsible for more than 80% of deaths in poor and developing nations. By 2020, it is predicted that over 178 million Indonesians aged 20 years and above will have DM, with a prevalence rate of 4.6% [4,5,6]. In Southeast Asia, DM affects 78.3 million people, with Indonesia ranked seventh (10 million DM patients) in the world after China, India, the United States, Brazil, Russia, and Mexico. These numbers are projected to keep increasing until 2040, when it is estimated that one in every ten individuals will have DM [7,8].

DM and its associated complications are responsible for the majority of deaths in many countries. Type 2 DM, also known as Non-Insulin Dependent DM (NIDDM), constitutes more than 90% of all diagnosed cases of diabetic in adults. The condition is characterized by a partial or complete inability to use insulin effectively (insulin resistance) despite the presence of functional insulin, thereby leading to hyperglycemia [9,10,11,12,13].

Although modern medications can be used to treat DM, either orally or injection, their high cost can pose a problem. In addition to these medications, traditional remedies derived from plants can also be used [14]. Squalene, a naturally occurring triterpenic hydrocarbon, is present in various natural products, such as shark liver and olive oils. It serves as a key intermediate in the biosynthesis of phytosterols and cholesterol in animals, plants, and humans [11]. It has been tested in pharmacological activity including antibacterial, anticancer, antiinflammatory, antidiabetic, and antioxidant properties. Studies on antidiabetic effects of squalene have been conducted, such as that by Farijati [15], who studied mixed squalene preparation (97.9%) and lecithin (2.1%). The result found that the area Under The Curve (AUC) of the treated group was smaller than the control. Widyawati et al. [16] stated the antihyperglycemic activity of squalene extracted from *Syzygium polyanthum* leaves in a diabetic rat model induced by Streptozotocin (STZ). This study was carried out by inhibiting the enzyme alpha glucosidase and increasing glucose uptake in muscles without affecting insulin secretion in pancreatic beta cells [11,16]. Based on previous studies, squalene has a broad antidiabetic effect capable of modulating glucose at the pancreatic tissue level and in peripheral tissues such as the liver, muscles, and adipose. Anti-diabetic drugs work through various mechanisms, such as stimulating insulin secretion, increasing insulin sensitivity, inhibiting the enzyme alpha-glucosidase and Dipeptidyl Peptidase IV (DPPIV), emulating a glucagon-like peptide-1 (GLP-1), and inhibiting sodium-glucose co-transporter-2 (SGLT-2). It shows there are still several possible mechanisms for squalene in DM.

Leptin, a hormone primarily secreted by adipose tissue, plays a significant role in weight regulation, reproductive function, and metabolic activity. Its main function is to modulate food intake and control long-term energy expenditure. However, it has also been found to stimulate appetite and increase food intake. Elevated levels of leptin in the blood have been linked to DM [17]. In obese individuals, higher plasma leptin levels are often observed as a consequence of increased body fat mass [18].

Free radicals are unstable chemical compounds that exist independently with unpaired electrons in the outer atomic or molecular orbitals, thereby making them highly reactive [19,20]. These chemical compounds can increase lipid peroxidation to malondialdehyde (MDA) and lower blood Sodium Dismutase (SOD). Free radicals are known to cause oxidative stress, resulting in cellular damage, including damage to pancreatic beta cells. Furthermore, this cell damage plays a significant role in the pathophysiology of diabetic and its complications. Antioxidants are compounds that can prevent oxidative stress by delaying its onset [4,5]. This has led to the hypothesis that supplementation with antioxidants provide chemical protection against the pathogenesis of diabetic and its complications [6]. A literature review shows that squalene has several pharmacological activity including hypolipidemic, hepatoprotective, cardioprotective, antitoxic, and antioxidant [21,22]. Its antioxidant properties are beneficial for skin diseases. However, to date, no study has investigated activity of squalene in a diabetic model.

The main objective of this study is to investigate the potential antidiabetic activity of squalene both in silico, using molecular docking analysis as well as in vivo, utilizing a Type 2 DM rat model (T2DM).

## 2. Results

### 2.1. In Silico Assay: Squalene Biological Prediction Activity by PASS Server

The biological activity of squalene is shown in Figure 1.

Figure 2 shows antidiabetic properties of squalene with the indigo constant Pa, used to predict biological activity. The gamma Pa values range from 0 to 1, where a value closer to 1 indicates a more accurate prediction. Squalene has a Pa value of 0.369, which suggests its potential as a treatment for diabetic neuropathy. A Pa value greater than 0.3 indicates that the prediction is based on computation and requires further validation in the laboratory. Additionally, squalene has been found to stimulate insulin expression, as indicated by a value of 0.354 in the insulin promoter. This suggests a correlation between squalene and insulin levels. Furthermore, Table 1 and Figure 3 show the target proteins of squalene.

The target of squalene is shown in Figure 3.

This in silico study found that squalene targeted on FNTA, RABGGTB, FNTB, SQUALENE, TRVP1, TRPV4, MAPK21, YWHAB, KRAS, RAF1, SOS1, HRAS, KRAS, NRAS, PIK31R1, PTPN1, SHC1, PIK3CA, CDK2, PRKDC, and KAT2B.

### 2.2. In Vivo Assay

#### 2.2.1. Effect of Squalene on Fasting Blood Glucose Level (FBGL)

Fasting BGL in diabetic rats on day 0, 3, 6, 9, 12, and 14 is shown in Table 2.

Table 2 showed that squalene has the ability to decrease FBGL from day 3 to 14, similar to the results observed with metformin, a standard oral antidiabetic drug capable of reducing FBGL. In contrast, diabetic control group showed consistently high FBGL from day 0 (285.20 ± 74.34 mg/dL) to day 14 (350.30 ± 159.88 mg/dL). Decreased FBGL by administering squalene by 58% while metformin reached 67%. There were significant differences among the groups on day 14 (*p* = 0.017), and post hoc tests showed that both squalene- and metformin-treated groups were significantly different from diabetic control at *p* values of 0.012 and 0.016, respectively.

#### 2.2.2. Effect of Squalene on DPPIV Level

DPPIV levels of each group are shown in Table 3.

DPPIV (Dipeptidyl peptidase-4) is an enzyme that plays a crucial role in regulating physiological processes such as digestion and eating. Therefore, it is also essential for managing the glucose levels in the body. This study revealed that the group with diabetic control had the highest DPPIV level (61.26 ± 15.06 ng/mL), while squalene-treated group had the lowest (44.09 ± 5.29 ng/mL).

#### 2.2.3. Effect of Squalene on Pancreas

Histopatology of pancreas in normal and diabetic rat is shown in Figure 4.

Figure 4A shows the normal structure of the pancreas, which is comprised of two main components, namely the PA and the IL. Alpha cells are situated peripherally within the IL, and exhibit small nuclei, dark, granulated cytoplasm (beta-cells), and a regular membrane. Beta cells, located in the central portion of the IL, have large nuclei and light, granulated cytoplasm. Figure 4B indicates the photomicrographs of the diabetic pancreas. The IL exhibited regular membrane and shrinkage, decreased number of cells, small nuclei, as well as dark and granulated cytoplasm (alpha cells) in the peripheral. Meanwhile, the central beta cells showed large nuclei, light, and granulated cytoplasm. Figure 4C shows the pancreas of the metformin-treated group, wherein the IL has irregular membrane with slightly increased cell numbers in the peripheral alpha cells as well as large nuclei with light, granulated cytoplasm in the central beta cells. Lastly, Figure 4D is a pictorial representation of the pancreas of squalene-treated individuals, which contains both PA and IL. The IL showed an irregular membrane with alpha cells exhibiting slightly increased cell numbers, small nuclei, dark, granulated cytoplasm in the periphery, and large nuclei with light, granulated cytoplasm in the central beta cells.

#### 2.2.4. Effect of Squalene on Leptin Levels

The leptin levels of each group are shown in Table 4.

According to the study leptin, a hormone produced by fat cells, which helps regulate body weight, has varying levels among diabetic control group (15.39 ± 1.77 ng/mL), squalene-treated group (13.86 ± 0.47 ng/mL), and the metformin-treated group (9.22 ± 0.84 ng/mL). The results showed that the leptin level in the metformin-treated group significantly lower different from squalene group and diabetic control (*p* < 0.05).

#### 2.2.5. Effect of Squalene on SOD Activity

SOD activity of each group can be seen on the Table 5 below.

This study found that SOD activity in diabetic control group (21.88 ± 0.97 U/mL) was lower than both squalene (22.47 ± 0.27 U/mL) and metformin (22.81 ± 0.08 U/mL). However, only the metformin-treated group showed a significant difference compared to diabetic control group (*p* = 0.029).

## 3. Discussion

Squalene is a triterpenic compound found in a large number of plants and other sources with a long tradition of research since it was first reported in 1926 [23,24,25]. In clinical and preclinical trials, squalene emulsion administration intravenously showed safe and well-tolerated, with a slower clearance from the circulation compared to plant sterol and triglycerides [26,27]. As adjuvant, this compound also showed a safe, effective, and preferred vaccine adjuvant [28,29]. Its bioactivities show some properties including cardioprotector, antibacterial, antifungal, anticancer, detoxifying, antioxidant, and antidiabetic [24,30,31]. Ebrahimi et al. [32] provide evidence that this compound can lead to clinical improvement of on SARS-CoV-2 patients. Research on the antidiabetic activity of squalene has been carried out previously both in silico and in vivo. Due to the presence of this compound in the active extract, fraction, and subfraction of plants such as *S. polyanthum* leaves extract. The in silico assay in this study used a DM-related protein receptor and method that was different from what had been carried out previously. Previous in vivo assay was conducted in type 1 DM animal model with STZ induction at high doses of 55 mg/kgbw with parameters of FBGL, insulin levels, weight, lipid profile, and immunohistochemistry. The novelty of this current research is to examine the squalene antidiabetic activity in type 2 DM rats with low doses of STZ induced in rats (30 mg/kgbw) that have received HFD which has not been conducted by other researchers.

Molecular docking is a type of bioinformatic model used to predict whether a compound is active before being tested [33]. This technique involves analysing protein-ligand interactions at the atomic level, which can be compared to the lock-and-key principle. Therefore, by studying these interactions, target structures for the active sites of proteins and gaining insights needed to determine the potential mechanisms of action for compounds can be discovered. Understanding protein-ligand interactions is crucial for comprehending biological regulation mechanisms, and it provides a theoretical foundation for designing and discovering new drug targets [34]. On the other hand, ligands can interact with proteins through various means, such as hydrogen bonding, hydrophobic bonding, van der Waals forces, and salt bridges, with their interaction characterized by binding affinity [35].

This study utilized molecular docking to explore the potential impact of squalene as a ligand on multiple targets, including TRPV1 gene and protein. TRPV1 is associated with the development of diabetic and is highly expressed in sensory nerve fibers in the pancreas. The depletion of TRPV1 may lead to elevated glucose intolerance in T2DM. Interestingly, the molecular docking results suggested that squalene potentially inhibits the expression of TRPV1. In addition, the study found that the expression of KARS protein was higher in diabetic patients compared to non-diabetics. Overexpression of KRAS protein was observed in cancerous samples of T2DM patients. The inhibition of KARS protein by squalene is beneficial for patients with T2DM. The findings also indicated that diabetic patients with a mutated KRAS gene had a higher risk of colon cancer than non-diabetics [36,37]. Moreover, squalene was found to stimulate insulin expression. The presence of squalene in the insulin promoter of 0.354 implies a correlation between squalene and insulin levels. Furthermore, Pa > 0.3 indicates that it can also be applied as an insulin promoter. This region (around 1000 bp) regulates the β cell-specific expression of insulin, and is controlled by both ubiquitous and pancreatic β cell-specific transcription factors [38]. This in silico analysis agrees with the findings of Mirmiranpour et al. [39] on the effect of squalene on lipid profiles in T2DM patients and Widyawati et al. [11] on squalene affinity against 2Q5S, 2HWQ, and 1FM9 receptors. These receptors are the proteins that regulate and control energy balance, biosynthesis of lipids, and adipogenesis.

Experimental animal models are categorized into two main types, namely spontaneous or genetically induced models, and induced or non-genetically-induced models. Spontaneous or genetic models refer to normal animals that exhibit phenotypic similarities to humans or abnormal animal species resulting from spontaneous mutations. Induced or non-genetic models, on the other hand, are animals whose normal physiological state is altered through interventions such as surgery, genetic manipulation, or chemical administration [40]. One of the chemicals used to induce experimental animals to become DM is STZ. STZ (2-deoxy-2-(3-methyl-3-nitrosourea)-1-D-glucopyranose), a cytotoxic glucose analogue, is a naturally occurring compound produced by the soil bacterium Streptomyces achromogenes and has a broad spectrum of antibacterial properties. After its discovery, it was used as a chemotherapeutic alkylating agent in the treatment of metastasizing pancreatic islet cell tumors and other malignancies [41,42,43,44]. STZ enters β-cells via the Glucose Transporter-2 (GLUT2) present in the plasma membrane of the β-cell and causes cell death by inducing DNA methylation through the STZ methylnitrosurea group. This triggers DNA damage, leading to pancreatic β-cell necrosis through depletion of cellular energy stores. The activation of Poly ADP Ribose Polymerase (PARP) in response to DNA damage further reduces cellular NAD+ [45]. STZ effects are visualized within 72 h after administration depending on the dose administered [33]. A single high dose of STZ can cause complete β-cell necrosis and diabetic within 48 h [46], making it a widely used method for creating an experimental model of Type 1 DM (T1DM) [41,47]. To create an animal model of type 2 DM, low doses of STZ were used concomitantly with nicotinamide or preceded by a high-fat diet. T1DM and T2DM are two distinct disease entities [48]. In T1DM, the immune system of patients attacks and destroys the insulin-producing pancreatic β cell. While on T2DM, the patient experiences high blood sugar, insulin resistance, and a relative lack of insulin. A high-fat diet is meant to introduce conditions such as insulin resistance that contribute to T2DM. In this study, rats with T2DM were administered an HFD before inducing low doses of STZ (30 mg/kgbw). This model differs from previous studies by Widyawati et al. [11], which induced experimental animals directly with larger doses (55 mg/kgbw) imitated by T1DM. This model is useful for evaluating the possibility of one of the mechanisms of action of squalene as an antidiabetic studied in this study through the parameters of DPPIV enzyme. Additionally, the presence of obesity induced by the HFD treatment may be associated with the parameter studied, namely leptin.

The administration of squalene decreased FBGL comparable to metformin, a conventional oral antidiabetic agent. The histopathological examination of the pancreas using H&E staining showed significant differences in the general histological organization of pancreatic tissue between normal and diabetic rats. This difference was evident due to the surrounding exocrine cells that were not affected by the induction [49]. Normal rats exhibited islets with regular membrane and normal appearance of alpha and beta cells, indicating no degenerated cells. This observation is in accordance with preliminary studies by Andrade-Cetto et al. [50]. and Juarez et al. [51]. In contrast, diabetic rats showed a significant reduction in islet volume and number, with a smaller size of diabetic control. Squalene-treated rats restored the histological appearance of the IL, with nearly regular and normal size. One limitation of the study was the lack of quantitative comparison between the size of each group. In addition, it did not examine insulin expression in beta cells, which indicates whether the beta cells were still functioning. The histopathological picture using H&E staining confirmed the difference between the normal and diabetic groups. Qualitatively, administration of squalene showed an improvement in IL size.

Dipeptidyl peptidase-4 (DPPIV) is an important enzyme that regulates insulin production by deactivating GLP-1, an incretin capable of regulating insulin secretion after eating [47,52,53]. Inhibiting this enzyme increases incretin levels, leading to prolonged post-prandial insulin action [54]. Therefore, DPPIV4 inhibitors exert glucose regulatory actions by prolonging the effects of GLP-1, ultimately increasing glucose-mediated insulin secretion and suppressing glucagon secretion [55]. DPPIV enzyme inhibitors are used to treat the diabetic conditions [56]. Rats developing insulin resistance due to HFD feeding showed more active GLP-1 and insulin in plasma. The same improved glucose tolerance with increased GLP-1 and leptin levels was found in DPPIV-depleted Dark Agouti rats with diet-induced obesity [57]. The present study found that DPPIV enzyme level in squalene group was lower than in diabetic control group, indicating a similar effect to the inhibition of DPPIV enzyme. However, the effect was not significantly different among groups, and a longer study period is needed to obtain a more profound effect.

Leptin is an adiposity hormone that plays an essential role in regulating energy homeostasis and glycemic control [58]. Nikmah and Dany [59] showed that the leptin levels of individuals with diabetes were higher than normal subjects. In this study, it was observed that leptin production significantly increased in diabetic conditions compared to metformin. The leptin levels in diabetic control group were higher than those in squalene and metformin-treated groups. These results suggested that BGL is controlled by giving squalene or metformin. This result supported the study by Shin et al. [60] and showed the serum concentrations of leptin and insulin were declined dramatically in metformin-treated standard chow and HFD obese rats. Metformin directly modulates adipocyte signaling by activating p44/p42 MAP kinase and impairs leptin secretion [61]. Liu et al. [62] reported that following squalene feeding for 4 weeks, body weight gain were lower in the squalene group. The levels of cholesterol, triglycerides, blood glucose, and leptin were significantly lower in squalene-treated rats. In the present study, treatment with squalene was given for only 2 weeks in HFD rats, although during this time squalene was able to reduce BGL significantly but not leptin levels. In a longer study period, this effect is likely to be be more pronounced. Petlola et al. [63] highlighted the high level of serum squalene are linked to visceral obesity and suggest that squalene in adipose tissue have destructive effects in abdominal obesity, thus suggesting it as one of metabolic syndrome markers. However, more study is needed to clarify the squalene’s effect on metabolic disorder.

STZ causes diabetic through multiple mechanisms, one of which is by increasing the production of Reactive Oxygen Species (ROS). This unstable molecule is widely used to determine the extent of organ injury, especially in diabetic conditions, as it causes lipid peroxidation and damage to the pancreas and other organs [64,65]. Superoxide Dismutase (SOD) are a group of enzymes that catalyze the dismutation of superoxide radicals (O_2_^−^) to molecular oxygen (O_2_) and hydrogen peroxide (H_2_O_2_), providing cellular defense against ROS [66]. The high level of SOD correlate with its antioxidant activity. Interestingly, this study found that SOD level were higher in both squalene and metformin than diabetic control. Additionally, ROS can lead to apoptosis by activating caspase 3 and stimulating the pro-apoptotic protein Bax while decreasing the anti-apoptotic protein Bcl2 [67]. The hyperglycemic conditions also lead to excessive calcium release from the sarcoplasmic reticulum, leading to apoptosis. In hyperglycemia condition, the excessive production of ROS lead to the release of pro-inflammatory cytokines such as IL-6, IL-1β, TNF alpha, and INFy. This can cause serious damage to pancreatic beta cells, resulting in a lack of insulin production. Previous studies have shown that squalene increases antioxidant status, including SOD, GST, GPx, CAT, and GSH, and reduce the lipid peroxidation marker (MDA) [68]. Squalene is also highly effective as radical scavanger on superoxide anion. Moreover, squalene stimulates the production of anti-inflammatory cytokines IL-10, IL-13, and IL-4, while downregulating the pro-inflammatory cytokines TNF-α and NF-κB [69,70]. In line with these findings, FBGL levels were significantly decreased in squalene-treated group compared to diabetic control, thereby indicating that squalene can act as an antihyperglycemic agent.

## 4. Materials and Methods

### 4.1. In silico Antidiabetic Analysis

Biological Activity analysis by using Pass server and Potential Target Identification (HIT PICT and SWISS TARGET PREDICTION) and Pathway Analysis (Protein-Protein Interaction) (STRINGDB).

### 4.2. Materials and Reagent

A high-fat diet comprising 60% fat, squalene, STZ, and tween 80 were purchased from Sigma-Aldrich (St. Louis, MO, USA) with 0.9% NaCl.

### 4.3. Animals

Healthy Wistar male rats weighing between 180–250 g, were obtained from the animal house of Universitas Sumatera Utara. The animals were acclimatized to room temperature and a 12-h light/dark cycle and were given free access to food and water for a week before the experimentation. The study was conducted in compliance with the regulations of the Animal Research Ethics Committee (AREEC), Faculty of Mathematics and Natural Sciences (FMIPA), Universitas Sumatera Utara (No. EC: 00521/KEPH-FMIPA/2020).

### 4.4. Diabetic Induction to High Fat Diet Rats

Normal rats were fed high-fat diets containing goat fat, wheat flour, cooking oil, duck egg yolks, and sugar for 30 days. After 16 h of fasting, STZ (30 mg/kg, dissolved in a 0.9% NaCl solution) was injected intra-peritoneally. Blood glucose concentration was measured using a glucometer (One Touch Basic) 72 h after the STZ injection to confirm the presence of diabetic. Rats with a BGL of over 200 mg/dL were selected and used to conduct this study [14].

### 4.5. Experimental Design

Diabetic rats (*n* = 30) were divided randomly into three groups (*n* = 10 of each). The first group was given Squalene (160 mg/kg), while the second group was administered metformin (45 mg/kg), and served as the positive control. The third group, which served as the negative control, was treated with normal saline (10 mL/kg). All treatments were prepared by dissolving them in NaCl 0.9% and tween 80 5% prior to administration.

### 4.6. DPPV Level Measurement

DPPV level of rats were measured using Rat DPPIV Elisa Kit (^®^Elabscience, Houston, TX, USA) using the manufacture guideline.

### 4.7. Histopathology of Pancreas

After sacrificing them with carbogen gas (95% O_2_ and 5% CO_2_), the pancreas was removed from the rats and subsequently fixed in 10% buffered formaldehyde for 2 h. After fixation, the pancreas was dehydrated with 70% alcohol for 60 min, 96% alcohol for 45 min, and absolute alcohol for 24 h. The samples were then cleaned with repeated xylene immersions and infiltrated with paraffin wax. Thermo Scientific STP 120-3 tissue processor automatically processed the samples, while Thermo Scientific Microm EC 350-1 modular tissue embedding center prepared the paraffin embedding. The paraffin-embedded tissues were sectioned into 5 µm slices using the Leica RM 125RTS microtome and placed on microscope slides. These slices were then stained with hematoxylin and eosin using the H&E staining method, and mounted with a cover slide and DPX mounting media [61] for further analysis [71].

### 4.8. Leptin Level Measurement

Leptin levels of rats were measured using Rat LEP (Leptin) Elisa kit (^®^Elabscience, USA) and prepared in accordance with the manufacture guideline.

### 4.9. SOD Activity Measurement

SOD activity of rats was measured using Total SOD (T-SOD) Activity Assay Kit (WST-1 Method) (^®^Elabscience, USA) and was prepared in accordance with the manufacture guideline.

### 4.10. Data Analysis

Data were expressed as mean ± standard error of the mean (S.E.M). The results were analysed using One-Way Anova followed by Games Howell (parametric) or Kruskall–Wallis Tets using the Mann–Whitney U Test (non parametric). A *p*-value of less than 0.05 was considered statistically significant.

Schematic representation of this study can be seen in Figure 5 below.

## 5. Conclusions

In conclusion, in silico assay conducted in this study provided evidence of antidiabetic activity of squalene by demonstrating its interaction with TRPV1 and KARS protein. In addition, in vivo analysis T2DM rat model showed that squalene exhibited antidiabetic activity by inhibiting the DPP IV enzyme and improving the size of IL. Furthermore, it played a role in regulating leptin and enhancing SOD activity.

## Figures and Tables

**Figure 1 molecules-28-03783-f001:**
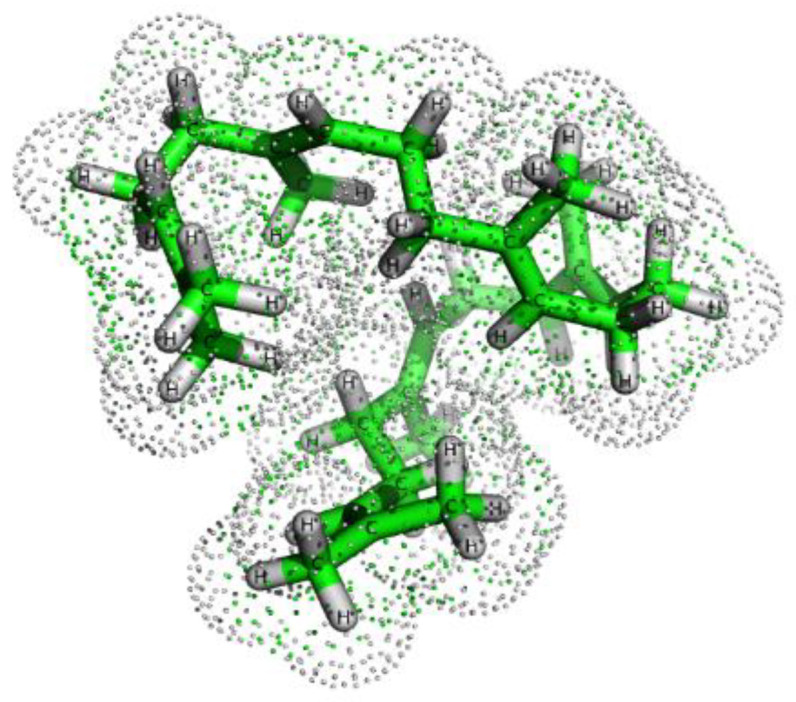
SD structure of squalene.

**Figure 2 molecules-28-03783-f002:**
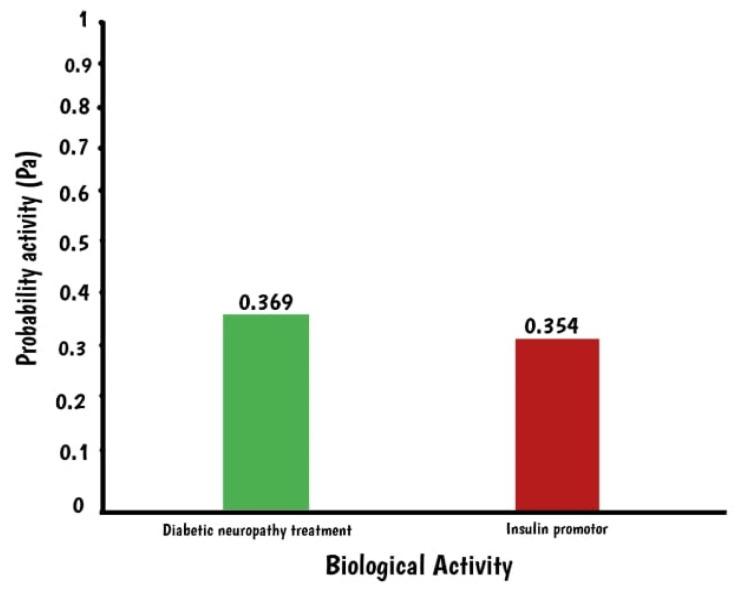
Biological activity of squalene.

**Figure 3 molecules-28-03783-f003:**
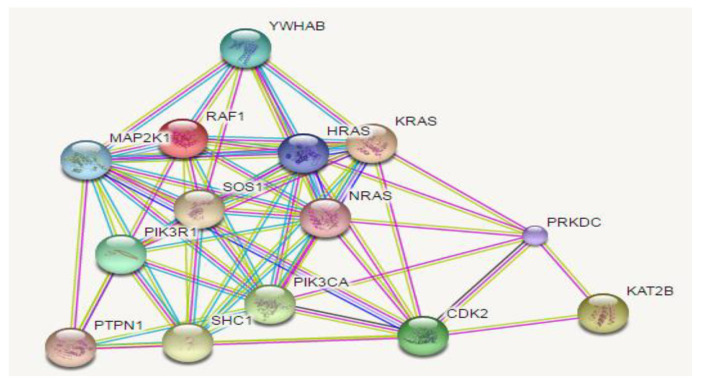
The target of squalene.

**Figure 4 molecules-28-03783-f004:**
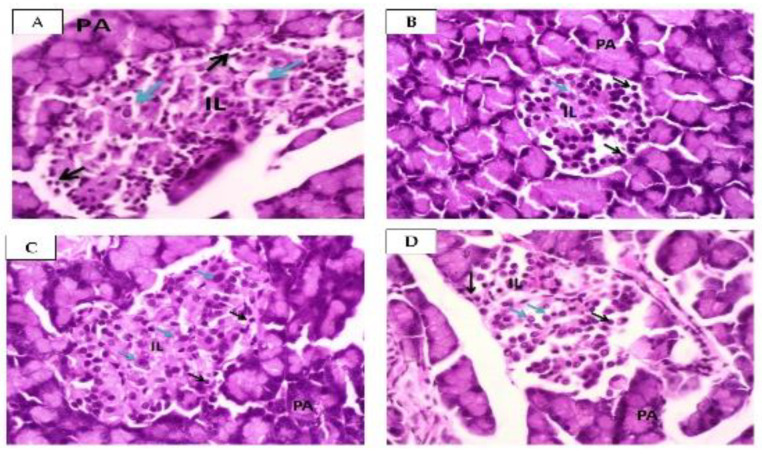
Histopathology of pancreas. ((**A**): Normal group, (**B**): Diabetic control, (**C**): Metformin, (**D**): Squalene; PA: Pancreatic asinus; IL: Islet of Langerhans; Black arrow: Alpha-cells; Blue arrow: Beta-cells; H&E staining, 400× magnification).

**Figure 5 molecules-28-03783-f005:**
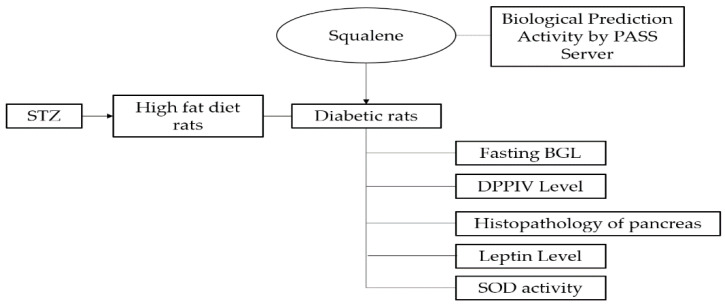
Schematic representation of analysis of antidiabetic activity of squalene.

**Table 1 molecules-28-03783-t001:** Target protein of squalene.

Active Compound	Potential Target
Squalene	FNTA
	KRAS
	RABGGTB
	RABGGTA
	FNTB
	SQUALENELE
	TRPV1
	TRPV4

**Table 2 molecules-28-03783-t002:** Effect of squalene on FBGL.

Group	FBGL (mg/dL) Mean ± SD					
	Day 0	Day 3	Day 6	Day 9	Day 12	Day 14
Squalene (*n* = 6)	318.40 ± 46.03	292.60 ± 43.13	262.80 ± 4.93	220.40 ± 36.15	175.00 ± 23.49	134.40 ± 16.95 ^a^*
Metformin (*n* = 6)	338.00 ± 25.13	302.40 ± 27.5	260.00 ± 26.89	215.40 ± 32.27	170.40 ± 40.31	113.18 ± 33.03 ^b^*
Diabetic control (*n* = 5)	285.20 ± 74.34	294.20 ± 89.90	294.40 ± 104.36	311.00 ± 118.34	319.20 ± 127.70	350.30 ± 159.88
*p*	0.302	0.962	0.676	0.399	0.145	0.017

(Data were analysed using Kruskall–Wallis followed by Mann–Whitney; ^a^* *p* = 0.028; ^b^* *p* = 0.016, compared to diabetic control).

**Table 3 molecules-28-03783-t003:** Effect of squalene on DPPIV level.

Group	DPPIV (ng/mL) Mean ± SD
Squalene (*n* = 6)	44.09 ± 5.29
Metformin (*n* = 6)	59.09 ± 8.10
Diabetic control (*n* = 5)	61.26 ± 15.06
*p*	0.105

(Data were analysed using One-way ANOVA).

**Table 4 molecules-28-03783-t004:** Effect of squalene on leptin level.

Group	Leptin (ng/mL) Mean ± SD
Squalene (*n* = 5)	13.86 ± 0.47 *^a^
Metformin (*n* = 4)	9.22 ± 0.84
Diabetic control (*n* = 5)	15.39 ± 1.77 *^b^
*p*	0.011

(Data were analysed using One-Way ANOVA followed by Games Howell; *^a^ *p* = 0.012; *^b^ *p* = 0.049 compared to Metformin).

**Table 5 molecules-28-03783-t005:** Effect of squalene on SOD activity.

Group	SOD (U/mL) Mean ± SD
Squalene (*n* = 6)	22.42 ± 0.27
Metformin (*n* = 6)	22.81 ± 0.08 *
Diabetic control (*n* = 4)	21.88 ± 0.97
*p*	0.036

(Data were analysed using Kruskal–Wallis Test followed by Mann–Whitney Test; * *p* = 0.029 compared to diabetic control).

## Data Availability

Not Applicable.

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
