# Peer review of "Analysis of Antidiabetic Activity of Squalene via In Silico and In Vivo Assay"

_molecules, 2023, doi:10.3390/molecules28093783_

Round 1

Reviewer 1 Report

Major Comments:

1. Are there controversies in this field? What are the most recent and important achievements in the field? In my opinion, answers to these questions should be emphasized. Perhaps, in some cases, novelty of the recent achievements should be highlighted by indicating the year of publication in the text of the manuscript.

2. The results and discussion section is very weak and no emphasis is given on the discussion of the results like why certain effects are coming in to existence and what could be the possible reason behind them?

3. Conclusion: not properly written.

4. Results and conclusion: The section devoted to the explanation of the results suffers from the same problems revealed so far. Your storyline in the results section (and conclusion) is hard to follow. Moreover, the conclusions reached are really far from what one can infer from the empirical results.

5. The discussion should be rather organized around arguments avoiding simply describing details without providing much meaning. A real discussion should also link the findings of the study to theory and/or literature.

6. Spacing, punctuation marks, grammar, and spelling errors should be reviewed thoroughly. I found so many typos throughout the manuscript.

7. English is modest. Therefore, the authors need to improve their writing style. In addition, the whole manuscript needs to be checked by native English speakers.

Reviewer 2 Report

The manuscript reports analysis of antidiabetic activity of squalene via in silico and in vivo assay. DM is a metabolic chronic disease on a global scale. Squalene for DM care was studied in this article. Fasting blood glucose level (FBGL), superoxide dismutase (SOD), leptin, and dipeptidyl peptidase IV (DPPIV) were studied to analysis the antidiabetic activities. Thus, I do not have any objection for publication of this article in Molecules. However, I have a few comments, which are listed below. Revision is essential for the publication.

1. Group squalene, metformin and diabetic control were studied in this article. However, the advantages of squalene over metformin have not been demonstrated.

2. The side effect of squalene should be discussed.

3. Animal experiments were carried out. It is better if in vitro cell experiment could be carried out to verified the result.

4. Did each group contain 10 rats. It should be clear in the manuscript.

5. In Figure 2, did diabetic neuropathy treatment and insulin promoter have connections. In addition, it should be indicated that Pa >0.3 is also applicable to insulin promoter.

6. The quality of Figure 2 should be improved.

Author Response

Dear Reviewer,

Round 2

Reviewer 1 Report

Authors addressed all of my comments. The revised manuscript can be accepted for final publication.